# RalA, PLD and mTORC1 Are Required for Kinase-Independent Pathways in DGKβ-Induced Neurite Outgrowth

**DOI:** 10.3390/biom11121814

**Published:** 2021-12-02

**Authors:** Takuya Kano, Ryosuke Tsumagari, Akio Nakashima, Ushio Kikkawa, Shuji Ueda, Minoru Yamanoue, Nobuyuki Takei, Yasuhito Shirai

**Affiliations:** 1Department of Applied Chemistry in Bioscience, Graduate School of Agricultural Sciences, Kobe University, Kobe 657-8501, Japan; soc0813t.k@gmail.com (T.K.); gyama_rin5334@yahoo.co.jp (R.T.); uedas@people.kobe-u.ac.jp (S.U.); yamanoue@kobe-u.ac.jp (M.Y.); 2Division of Signal Functions, Biosignal Research Center, Kobe University, Kobe 657-8501, Japan; anakashima@peerson.kobe-u.ac.jp (A.N.); ukikkawa@kobe-u.ac.jp (U.K.); 3Department of Bioresource Science, Graduate School of Agricultural Science, Kobe University, Kobe 657-8501, Japan; 4Department of Brain Tumor Biology, Brain Research Institute, Niigata University, Niigata 951-8585, Japan; nobtak@bri.niigata-u.ac.jp

**Keywords:** diacylglycerol kinase, phosphatidic acid, mammalian target of rapamycin, RalA, phospholipase D (PLD), neurite

## Abstract

Diacylglycerol kinase β (DGKβ) is an enzyme that converts diacylglycerol to phosphatidic acid and is mainly expressed in the cerebral cortex, hippocampus and striatum. We previously reported that DGKβ induces neurite outgrowth and spinogenesis, contributing to higher brain functions, including emotion and memory. To elucidate the mechanisms involved in neuronal development by DGKβ, we investigated the importance of DGKβ activity in the induction of neurite outgrowth using human neuroblastoma SH-SY5Y cells. Interestingly, both wild-type DGKβ and the kinase-negative (KN) mutant partially induced neurite outgrowth, and these functions shared a common pathway via the activation of mammalian target of rapamycin complex 1 (mTORC1). In addition, we found that DGKβ interacted with the small GTPase RalA and that siRNA against RalA and phospholipase D (PLD) inhibitor treatments abolished DGKβKN-induced neurite outgrowth. These results indicate that binding of RalA and activation of PLD and mTORC1 are involved in DGKβKN-induced neurite outgrowth. Taken together with our previous reports, mTORC1 is a key molecule in both kinase-dependent and kinase-independent pathways of DGKβ-mediated neurite outgrowth, which is important for higher brain functions.

## 1. Introduction

Diacylglycerol kinase (DGK) phosphorylates diacylglycerol (DG) to produce phosphatidic acid (PA). This enzyme is considered to play an important role in signal transduction by regulating the balance between two lipid messengers since DG is an activator of protein kinase C (PKC) and PA can regulate the functions of several enzymes, including mTOR [1]. Indeed, of the 10 mammalian DGK subtypes [2,3,4], DGKα is necessary for anergy [5,6], and DGKδ is involved in diabetes [7].

The β subtype of DGK (DGKβ) is abundantly expressed in neurons, especially in the cerebral cortex, hippocampus and striatum, and its expression increases on the postnatal day when neural networks are formed [8]. We previously reported that DGKβ induces the formation of neural networks, including neurite outgrowth and spinogenesis, and DGKβ knockout (KO) mice showed impairments in emotion and memory [9,10,11,12]. We also found that a unique localization of DGKβ at the plasma membrane was critical for its functions. DGKβ consists of regulatory domains at the N-terminus and kinase domains at the C-terminus [13]. In the regulatory domain, EF-hands are responsible for calcium binding, and the C1 domain is homologous to PKC. The C1 domain is necessary for plasma membrane localization and the ability to induce neurite outgrowth of DGKβ [13]. Furthermore, we have recently found that DGKβ induces neurite outgrowth and spinogenesis by activating mTORC1 in a kinase-dependent pathway [14]. Surprisingly, we found that the kinase-negative (KN) mutant also induces neurite outgrowth in SH-SY5Y cells in a series of experiments to explore the mechanism of DGKβ-induced neurite outgrowth. However, the precise molecular mechanisms of DGKβKN-induced neurite outgrowth are unknown. Therefore, we explored the molecular mechanism underlying DGKβKN-induced neurite outgrowth, focusing on the binding proteins. Finally, we found that RalA binds to the C1 domain of DGKβ and that activation of PLD and mTORC1 is important for DGKβKN-induced neurite outgrowth.

## 2. Materials and Methods

### 2.1. Materials

We used the following compounds and materials: rapamycin and FIPI (Merck, Darmstadt, Germany), peroxidase-conjugated Affinipure goat anti-rabbit/mouse IgG, and Alexa Fluor 594 (Alexa594)-conjugated goat anti-mouse IgG (Jackson Immunoresearch Inc., West Grove, PA, USA). We also used the following cell culture reagents: Fugene^®^ HD (Promega, Madison, WI, USA), D-MEM medium, Ham’s F-12 medium, D-MEM/Ham’s F-12 medium, penicillin/streptomycin, Opti-MEM, HBSS, Glutamax and Nerve-Cell Culture System (FUJIFILM Wako Pure Chemical Corporation, Osaka, Japan), and FBS (Thermo Fisher Scientific, Waltham, MA, USA). siRNA for RalA, CGCGGTGCAGATTCTTCTTAA, was obtained from QIAGEN (Venlo, The Netherlands). The plasmids of DGKβ and their mutants were previously described [13], and the plasmids of RalA and their mutants were previously described [14]. Mouse anti-FLAG was donated by Dr. Saito (Biosignal Research Center, Kobe Univ., Kobe, Japan), and rabbit anti-PSD95 was donated by Dr. Fukada (National Institute for Physiological Sciences, Aichi, Japan).

### 2.2. Cell Culture

COS-7 cells were cultured in DMEM with 10% FBS, and SH-SY5Y cells were cultured in DMEM/Ham’s F-12 medium with 15% FBS and 2 mM GlutaMAX at 37 °C in a humidified atmosphere containing 5% CO_2_. All media were supplemented with penicillin (500 units/mL) and streptomycin (500 mg/mL).

### 2.3. Neurite Outgrowth Evaluation

SH-SY5Y cells were plated onto glass bottom dishes at 1.0 × 10^5^ cells/dish. After 24 h, the medium was removed, and the cells were transfected with plasmids by lipofection using Fugene^®^ HD Transfection Reagent. After 4 h, the medium was added, and the cells were further cultured for 48 h. Each compound was added to the medium for 24 h. After culturing, the cells were fixed with 4% PFA and observed under confocal microscopy.

### 2.4. Pull Down Assay

COS-7 cells were separately transfected with plasmids encoding FLAG-RalA (or the mutants) or GFP-DGKβ (or the mutants) by electroporation (Gene Pulser II, Bio-Rad Laboratories, Hercules, CA, USA). After 48 h, the cells were harvested and lysed in homogenate buffer. Total lysate was mixed with anti-FLAG antibody in PBS-T containing 1% bovine serum albumin for 16 h at 4 °C. Then, the mixtures were immunoprecipitated by protein G Sepharose (GE Health Care, Boston, MA, USA) and subjected to immunoblotting.

### 2.5. Immunoblotting

SH-SY5Y cells were transfected with plasmids by lipofection and cultured for 48 h. The cells were treated with rapamycin and/or serum-free medium for 24 h. The cells were harvested and homogenized in ice-cold homogenate buffer (1 mM EGTA, 1 mM EDTA, 20 mM Tris-HCl, 1 mM MgCl_2_, 1 mM PMSF, 1% Triton X-100, pH 7.4) using handy sonic (UR-20, TOMY SEIKO Co., Ltd., Tokyo, Japan). After centrifugation, the lysates were obtained. Immunoblotting was performed as described previously [13]. Briefly, the samples were subjected to 10% SDS–PAGE, followed by blotting onto a polyvinylidene difluoride membrane (Millipore, Darmstadt, Germany). Nonspecific binding sites were blocked by incubation with 5% skim milk in ×1 concentrated phosphate buffered saline (PBS) containing 0.03% Triton X-100 (PBS-T) for 1 h. The membrane was incubated with the appropriate antibody for 1 h at room temperature. After washing with PBS-T, the membrane was incubated with peroxidase-labeled anti-rabbit/mouse IgG for 30 min. After three rinses with PBS-T, the immunoreactive bands were visualized using an Immunostar (Wako, Osaka, Japan). The density of the bands was analyzed by ImageJ. To detect phosphorylated protein, we used 5% BSA for blocking instead of skim milk, and ×1 concentrated Tris-buffered saline (TBS), containing 0.03% Tween 20 (TBS-T) instead of PBS-T.

### 2.6. Subcellular Localization of DGKβ, Mutants and RalA

CHO-K1 cells were plated onto glass bottom dishes at 1.0 × 10^5^ cells/dish. After 24 h, the cells were transfected with plasmids encoding FLAG-RalA and GFP-DGKβ or the mutants by lipofection using Fugene^®^ HD Transfection Reagent. After culturing for 24 h, the cells were fixed with 4% PFA. The cells were Tritonized for 30 min and then blocked with 10% normal goat serum for 2 h. The cells were incubated with anti-FLAG antibody for 16 h at 4 °C and then visualized with Alexa 594-labeled goat anti-mouse IgG, followed by observation under confocal microscopy.

## 3. Results

Our previous results suggested that the kinase-negative (KN) mutant of DGKβ also induced neurite outgrowth in SH-SY5Y cells, although the effect was weaker than wild-type (WT) DGKβ [13]. Therefore, we precisely compared neurite outgrowth ability between WT and KN in this study. As shown in Figure 1A, both WT and KN DGKβ were localized on the plasma membrane of SHSY-5Y cells. Figure 1B shows that approximately 55% of SHSY-5Y cells expressing GFP did not have any neurites (white column), and 12% of the cells showed multiple neurites with several branches (gray column). After overexpression of GFP-DGKβ, the percentage of cells with no neurites reached 23%, and that of several neurites increased to 45%. The ratio of the cells having several neurites to that of cells with no neurites was approximately 0.22 in the case of the control. In contrast, the ratio dramatically increased to approximately 1.96, clearly indicating the ability of WT DGKβ to induce neurites. In the case of GFP-DGKβKN, the percentage of cells with no neurites and several neurites were 31% and 35%, respectively. Although the decrease in the number of cells with no neurites and the increase in the number of cells with multiple neurites were less than those of the WT, the differences between the control and KN groups were significant (Figure 1B). The ratio of cells with several neurites to cells with no neurites was 1.13. These results indicated that DGKβKN also induced neurite outgrowth, although it was less effective than the WT.

To investigate the mechanism of DGKβKN-induced neurite outgrowth, we focused on the binding protein(s). First, we tried to identify the binding proteins by pull-down assay using mass spectrometry. However, we could not confirm direct binding to GFP-DGKβ, although some candidate proteins were found. In previous experiments, we found that a small GTPase, RalA, bound to the C1 domain of PKCη, which is similar to the C1 domain of DGKβ [14]. Therefore, we tested the binding of RalA to DGKβ and found that RalA bound to DGKβ (Figure 2). Hence, we examined the interaction between DGKβKN and RalA by pull down assay and found that RalA also bound to DGKβKN, although the amount was slightly less than that of WT (Figure 2). The interaction of DGKβ and DGKβKN was consistent with their colocalization and FLAG-RalA at the plasma membrane (Figure 3). Next, to identify the binding domain, we performed pull-down assays using C1A or C1B domain-deleted mutants of DGKβ based on the fact that the DGKβ C1 domain consists of two subdomains, C1A and C1B. As shown in Figure 2, the binding of the mutants to RalA was dramatically reduced by mutations in C1A or C1B, indicating that the entire C1 domain is important for binding to RalA. The membrane localization of the C1A mutant disappeared, and the C1B mutant showed very faint plasma membrane localization (Figure 3). These results indicate that RalA binds to DGKβ via the C1 domain and that binding is important for the membrane localization of DGKβ.

It is known that membrane localization of DGKβ is necessary for the induction of neurite outgrowth; the C1A and C1B mutants lose the ability to induce neurites [13], suggesting that binding to RalA may be a key for DGKβKN-induced neurite outgrowth. Therefore, to confirm that RalA was involved in DGKβKN-induced neurite outgrowth, we performed knockdown experiments using a siRNA targeting RalA. As expected, siRNA treatment against RalA significantly inhibited DGKβKN-induced neurite outgrowth (Figure 4A,B). In the case of siRNA treatment, the ratio of the cells having several neurites to that of cells with no neurites was 0.33, similar to the control (0.22). These results indicated that the interaction with RalA is necessary for its neurite induction ability. Furthermore, to confirm that the activation of RalA is important for DGKβ-induced neurite induction, we compared the binding of DGKβ to the active form of RalA (G23V) or negative form (S28N). As shown in Figure 4C, DGKβ WT bound to the active form of RalA more than the negative form. Importantly, DGKβ also bound to the active form of RalA, although the amount was smaller than that of WT.

It is well known that RalA binds to many proteins, including PLD and calmodulin [15]. We hypothesized that PLD is also involved in DGKβ KN-induced neurite outgrowth because PLD and DGKβ produce PA, which contributes to neurite outgrowth [16] and activates mTORC1 in the pathway by which DGKβ induces neurites [17,18]. To investigate the involvement of PLD in DGKβKN-induced neurite outgrowth, we tested the PLD inhibitors ethanol and FIPI. Ethanol inhibited DGKβKN-induced neurite outgrowth (Figure 5), and FIPI also showed inhibition, indicating that PLD is involved in DGKβKN-induced neurite outgrowth. These results suggested that DGKβKN-induced neurite outgrowth is mediated by the activation of PLD, suggesting that PA production is important for DGKβ KN-induced neurite outgrowth. In addition, we previously reported that rapamycin, an mTORC1 inhibitor, completely abolished WT DGKβ-induced neurite outgrowth, suggesting that mTORC1 is also involved in DGKβKN-induced neurite outgrowth. To confirm this hypothesis, we investigated the effect of rapamycin on DGKβKN-induced neurite outgrowth. When rapamycin was used to treat cells overexpressing DGKβKN, the ratio of cells with several neurites to cells with no neurites (the neurite induction index) became 0.30 (Figure 6A), which was almost the same as the index when rapamycin was used to treat cells overexpressing DGKβ (0.29). The inhibitory effect of rapamycin suggested the involvement of mTORC1 in DGKβKN-induced neurite induction. Furthermore, we examined phosphorylation of the S6 protein, a downstream molecule of mTORC1. mTORC1 promotes protein synthesis by phosphorylation of p70S6 kinases and eukaryotic initiation factor 4E (eIF4E)-binding proteins (4E-BPs), and subsequently, the p70S6 kinase phosphorylates the S6 protein. Therefore, we investigated whether DGKβKN overexpression increased S6 protein phosphorylation by Western blotting using S6 and phosphor-S6 antibodies. Figure 6B shows that overexpression of DGKβKN resulted in phosphorylation of the S6 protein, although it was slightly weaker than that in the case of DGKβ overexpression, and that both S6 phosphorylation by DGKβWT and KN were abolished by rapamycin treatment. These results indicated that mTORC1 also contributes to DGKβKN-induced neurite outgrowth.

## 4. Discussion & Conclusions

We showed for the first time that DGKβ induced neurite outgrowth through a kinase-independent pathway involving RalA, PLD and mTORC1. Previously, we also showed that activation of mTORC1 by DGKβ-produced PA was involved in DGKβ-induced neurite outgrowth [15]. In other words, there are kinase-dependent and kinase-independent pathways in DGKβ-induced neurite outgrowth, and mTORC1 is important for both pathways. Notably, there are two kinds of PA production by DGKβ and PLD that activate mTORC1, suggesting the importance of neurite outgrowth by mTORC1. Indeed, recent studies have revealed that mTORs induce the formation of neural networks, including dendrites and spines [19,20] and that mTORs are also involved in translation and actin cytoskeleton dynamics, which are important for neurite outgrowth [21,22].

Ral is a member of the Ras family of small GTPases and has been shown to be involved in neurite branching [23]. This finding is reflected in the morphological properties of neurons when DGKβ is overexpressed; DGKβ overexpression induces branching and spinogenesis [10,13,17]. Considering that branching and spinogenesis accompany membrane extension, the regulation of membrane trafficking by RalA, among the multiple functions of RalA, may be related to DGKβ-induced morphological changes. In addition, RalA has been reported to be involved in the polarity of neurons [24]. These studies support the important function of RalA even in DGKβ-induced morphological changes. Interestingly, Sec proteins, including Sec 5, which is one of the binding proteins of RalA, were involved in the regulation of membrane trafficking and neuronal polarity by RalA [23,24]. Sec 5 is also reported to regulate mTORC1 [25,26]. Accordingly, not only PLD but also Sec 5 may be involved in DGKβ-induced neurite outgrowth. On the one hand, Ral proteins consist of two proteins, RalA and RalB, which are 85% identical. However, we did not determine whether RalB binds to DGKβ. Although we speculate that RalB does not bind to the C1 domain of DGKβ because RalB does not bind to the C1 domain of PKCη [14], the possibility that RalB is involved in DGKβ-induced neurite outgrowth cannot be ruled out because a relationship between RalB and mTORC1 has also been reported [25,26].

In our previous paper, we showed that the type II DGK inhibitor R59949 decreased DGKβ-induced neurite outgrowth to the control level [15]. This result was controversial because DGKβKN partially but significantly induced neurites, as shown in this work. The controversial results suggest that R59949 has some effect in addition to abolishing the kinase activity of DGKβ. Therefore, we examined the effect of R59949 on the interaction between DGKβKN and RalA. We found that R59949 significantly inhibited the interaction between DGKβKN and RalA in a dose-dependent manner (Figure 7). This result indicated that R59949 inhibits not only the kinase activity of DGKβ but also the interaction with RalA. The inhibition seems to be due to the entire conformational change of DGKβ because R59949 binds to the catalytic domain of DGK [27], while RalA interacts with the C1 domain.

In summary, in addition to the kinase-dependent pathway in which DGKβ produces PA from DG and consequently activates mTORC1, DGKβ interacts with RalA via the C1 domain at the plasma membrane, followed by activation of PLD. PLD also produces PA and activates mTORC1. In other words, DGKβ activates PA-mTORC1 signaling both in kinase-independent and kinase-dependent pathways, contributing to neurite outgrowth induction.

## Figures and Tables

**Figure 1 biomolecules-11-01814-f001:**
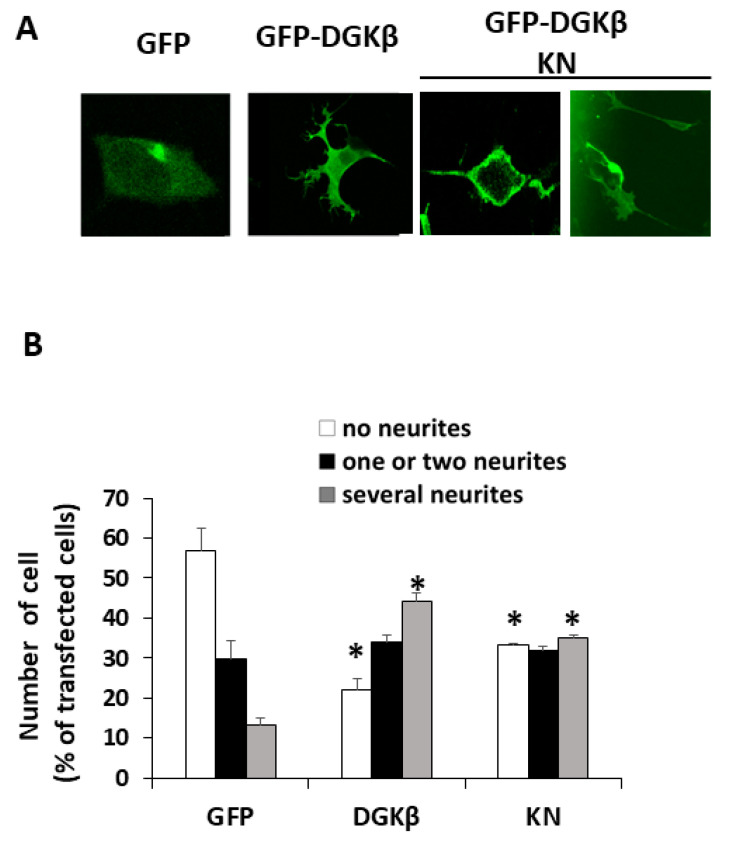
DGKβKN-induced neurite outgrowth in SH-SY5Y cells. (**A**) Typical images of SH-SY5Y cells overexpressing green fluorescent protein (GFP), GFP-DGKβ or GFP-DGKβKN. (**B**) Comparison of the number of cells having no (white column), one or two (black column) and several neurites (gray column) by overexpressing GFP (control), GFP-DGKβ (WT) or GFP-DGKβKN (KN). More than 100 cells were observed in each experiment and three independent experiments were performed. The mean and SEM of number of the cells are shown as percentage to the transfected cells. * *p* < 0.05 followed by Student’s *t*-test (vs. the respective controls expressing GFP alone).

**Figure 2 biomolecules-11-01814-f002:**
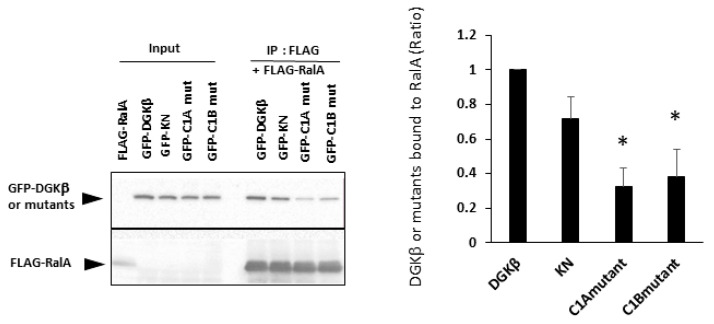
Binding of RalA to DGKβ via the C1 domain. Pull down assay was performed using COS-7 cells separately overexpressing FLAG-RalA and GFP-DGKβ, GFP-DGKβKN, C1A or C1B mutants. Three independent experiments were performed. The mean and SEM of density of bands are shown as ratio to the wild type (WT) DGKβ. * *p* < 0.05 followed by Student’s *t*-test (vs. GFP-DGKβ).

**Figure 3 biomolecules-11-01814-f003:**
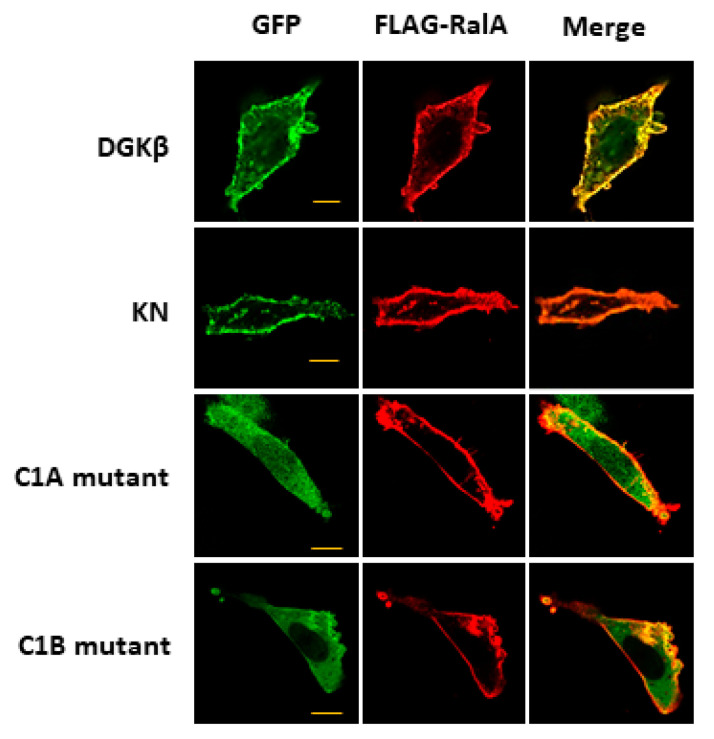
Colocalization of RalA with DGKβ on the plasma membrane. Typical images of SH-SY5Y cells overexpressing GFP-DGKβ, GFP-DGKβKN, GFP-DGKβ C1A mutant or GFP-DGKβ C1B mutant. Localization of GFP-DGKβ and the mutants were visualized based on GFP fluorescence (green), and FLAG-RalA localization was based on immunofluorescent staying using FLAG antibody and Alexa Fluor 594 (Alexa594)-conjugated goat anti-mouse IgG as secondary antibody (red).

**Figure 4 biomolecules-11-01814-f004:**
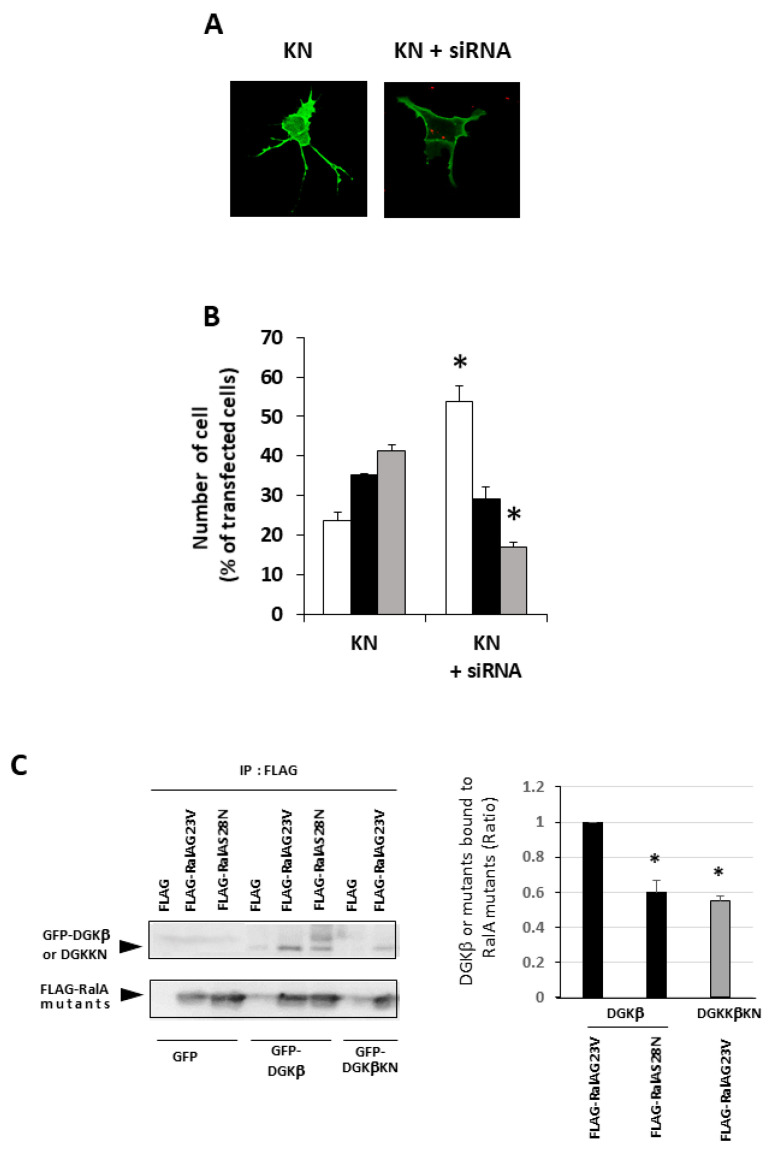
Importance of RalA in the DGKβKN-induced neurite outgrowth. (**A**) Typical image of SHSY-5Y overexpressing GFP-DGKβKN before and after treatment of siRNA for RalA. (**B**) Effect of siRNA for RalA on the number of cells having no (white column), one or two (black column) and several neurites (gray column). The plasmid of GFP-DGKβKN was transfected with or without 100 nM siRNA for RalA. More than 100 cells were observed in each experiment and three independent experiments were performed. The mean and SEM of the number of the cells are shown as percentage to the transfected cells. * *p* < 0.05, followed by Student’s *t*-test (vs. the control expressing GFP-DGKβ KN). (**C**) Active form of RalA bound to both DGKβ or DGKβKN. Pull down assay was performed using COS-7 cells separately overexpressing FLAG-RalA mutants and GFP-DGKβ or GFP-DGKβKN. Three independent experiments were performed. The mean and SEM of density of bands are shown as ratio to the WT DGKβ. * *p* < 0.05 followed by Student’s *t*-test (vs. GFP-DGKβ).

**Figure 5 biomolecules-11-01814-f005:**
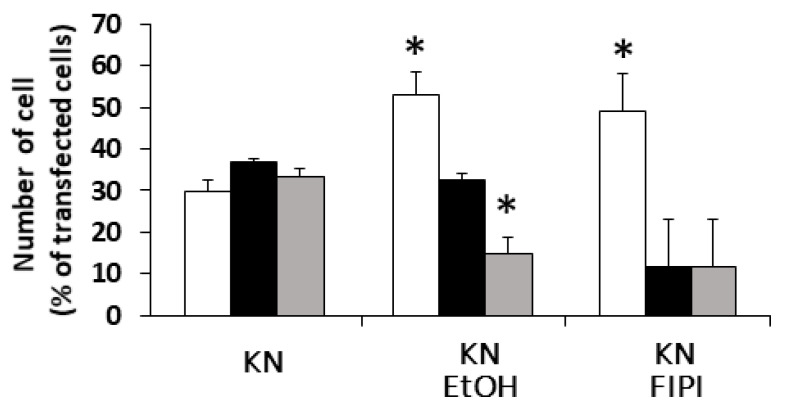
Effects of Phospholipase D (PLD) inhibitors on the DGKβKN-induced neurite induction. The plasmid of GFP-DGKβKN was transfected into SHSY-5Y cells. After 24 h, EtOH or FIPI was added to make a final concentration 0.3% and 1 μM respectively, and further cultured for 24 h. More than 100 cells were observed in each experiment and three independent experiments were performed. White, black and gray columns show percentage of the number of cells having no, one or two and several neurites, respectively. The mean and SEM of the number of the cells are shown as percentage to the transfected cells. * *p* < 0.05, followed by Student’s *t*-test (vs. the control expressing GFP-DGKβKN).

**Figure 6 biomolecules-11-01814-f006:**
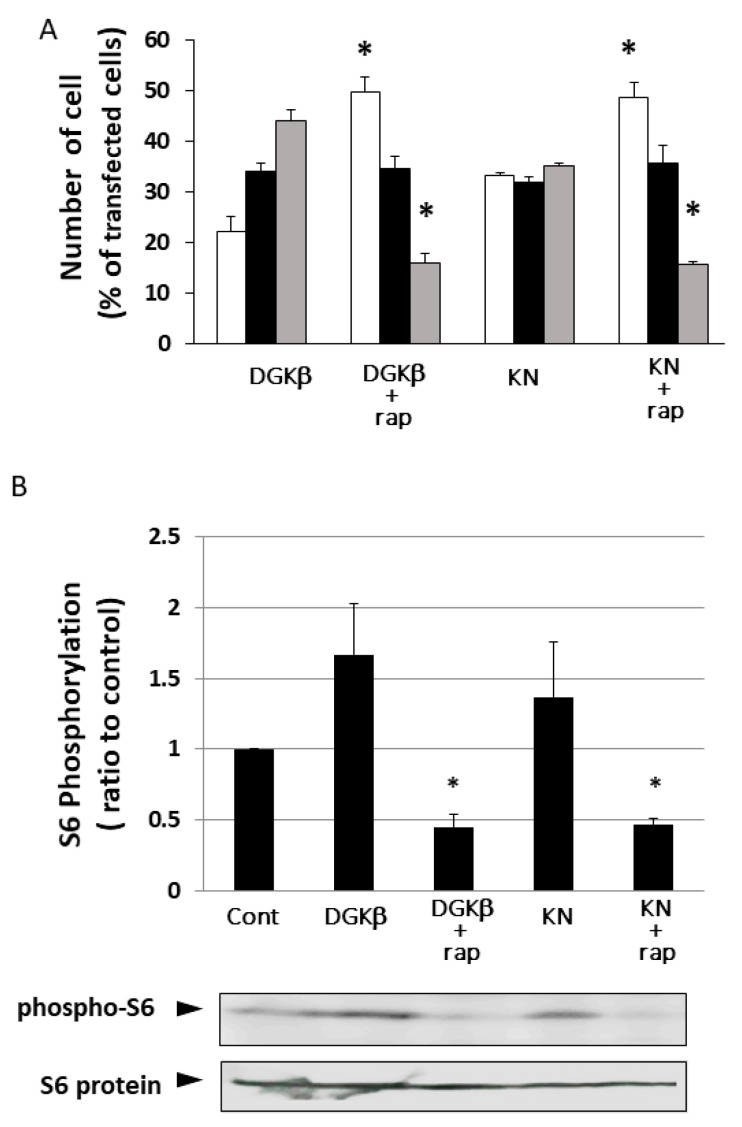
Involvement of mTORC1 in kinase-dependent and independent pathways for DGKβ-induced neurite outgrowth. (**A**) Effects of rapamycin on the DGKβ or DGKβKN-induced neurite outgrowth. The plasmid of GFP-DGKβ or GFP-DGKβKN was transfected into SHSY-5Y cells. After 24 h, 1 μM rapamycin was added and further cultured for 24 h. After fixed, more than 100 cells were observed in each experiment and three independent experiments were performed. White, black and gray columns show percentage of the number of cells having no, one or two and several neurites, respectively. The mean and SEM of number of the cells are shown as percentage to the transfected cells. * *p* < 0.05, followed by Student’s *t*-test (vs. the cells expressing GFP-DGKβ or GFP-DGKβKN respectively). (**B**) Phosphorylation level of S6 protein. SH-SY5Y cells overexpressing GFP-DGKβ or GFP-DGKβKN were cultured for 24 h, and then the cells were treated with 1 μM rapamycin for 24 h. The lysates from these cells were applied to immunoblotting and detected with phosho-S6 and S6 antibodies (bottom blots). Quantification analysis of S6 phosphorylation level. Ratios of phosphorylation to total protein were normalized by the ratio in the control (*n* = 3). * *p* < 0.05, followed by Student’s *t*-test (vs. control).

**Figure 7 biomolecules-11-01814-f007:**
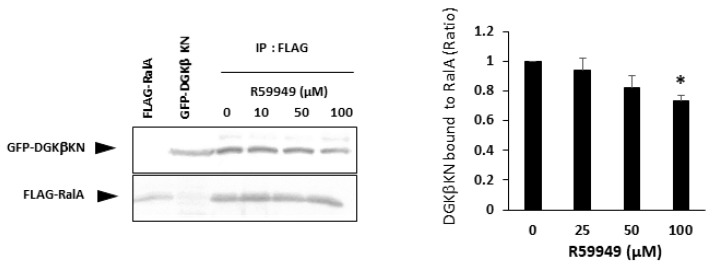
Effect of R59949 on the interaction between DGKβKN and RalA. Pull down assay was performed using COS-7 cells separately overexpressing FLAG-RalA mutants and GFP-DGKβKN in the presence of several concentrations of R59949. Three independent experiments were performed. The mean and SEM of density of bands are shown as the ratio to the WT DGKβ. * *p* < 0.05 followed by Student’s *t*-test (vs. GFP-DGKβ KN without R59949).

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
