# Peer review of "RalA, PLD and mTORC1 Are Required for Kinase-Independent Pathways in DGKβ-Induced Neurite Outgrowth"

_biomolecules, 2021, doi:10.3390/biom11121814_

Round 1

Reviewer 1 Report

In this paper, the authors followed up their previous finding that DGK-beta induces neurite outgrowth partially via mechanism(s) independent from its kinase activity. They provide convincing evidence that this kinase-independent process requires RalA, PLD and mTORC1. Overall, their conclusions are novel, important, and well supported by data. A weakness of this work, however, is the shortage of mechanistic insights. It does not necessarily diminish the value of their findings, and I therefore support the publication of this paper in principle; however, I suggest slightly weakening the statements addressing mechanisms, which are misleading in the current form (Major points 1 and 2). I also have a few concerns about how data are presented (Major points 3 and 4).

Major points:

  1. Their data shows RalA, PLD and mTORC1 are required for the neurite outgrowth induced by DGK-beta-KN. However, it does not necessarily mean that DGK-beta-KN activates the RalA-PLD-mTORC1 sequential pathway. They only observed mTORC1 activation (which itself is questionable- see Major point 4); there is no evidence for activation of RalA and PLD (they only show DGK-beta-KN binds RalA). Thus, they cannot exclude the possibility that DGK-beta-KN activates mTORC1 independent from the RalA-PLD pathway. Hence, saying ‘Sequential activation of RalA, PLD and mTORC1 is important for…’ in the title is misleading. I propose to change it to, for example, ‘RalA, PLD and mTORC1 are required for ….’.

  1. In the same vein, the abstract should be amended as ‘binding of RalA and activities of PLD and mTORC1 are involved…’ (l.23).

  1. Figure 2A and 2B: I am confused by input blots; why are the bait and prey proteins separately expressed there? These samples do not seem to come from the cells used for the IP experiments. These should be replaced with the real input blots (=pre-IP lysate of cells expressing both proteins), showing (supposedly) similar expression levels of prey proteins.

  1. Figure 6B: This data is questionable for two reasons. 1) Why are no blots shown? 2) Why does only this data lacks p-values? In particular, is ‘KN’ significantly different from ‘Cont’?

Minor points:

  1. (l. 31) ‘DGK is phosphorylates’: delete ‘is’.
  2. (l. 34) protein kinase C ‘(PKC)’: Introduce abbreviation.
  3. (l. 53) focusing ‘on’
  4. (l. 98 and 104) What does 0.01M PBS/TBS mean?
  5. (l. 141) RalA appears out of the blue here… why did authors try RalA? Is there any literature, or reason to suspect that it might bind DGK?
  6. (ll. 144-145) supported by: A mere colocalization does not support physical interaction. ‘consistent with’ is better here.
  7. (l. 152) Fig. 2B: this should be Fig. 3
  8. (l. 200) ‘the neurite induction index’: need definition/explanation.
  9. (l.242) worlds: words
  10. (l. 275) why ‘Indeed’? How should R59949, which binds to the catalytic domain, interrupt the RalA interaction, which occurs at the C1 domain?

Author Response

Thank you for constructive comments to make the manuscript better. According to your suggestions, we performed additional experiments and revised manuscript.

The revised sentence was in red. The followings are our responses to respective reviewer’s comments.

Comments to reviewer #1

1, According to your suggestion, the title was changed.

2, According to your suggestion, “sequential” was omitted on the line in the abstract.

  1. We performed additional experiments, we totally revised Fig. 2.
  2. We added the blot and p-values in Fig. 6B. Unfortunately, the difference between control and both WT and KN were not significant because of big error bar. That’s why we didn’t use “significant “ in the text.

Minor

  1. “is” was deleted.
  2. (PKC) was added.
  3. “on” was added.
  4. I described what PBS and TBS are.
  5. We added description why we finally tried RalA.
  6. We revised as you suggested.
  7. I changed Fig.2B to Fig. 3.
  8. I gave a definition in the text.
  9. Revised
  10. We revised to be understandable.

Reviewer 2 Report

The present article is a follow up of previous studies by the authors regarding the contribution of DGKbeta to neurite spine formation. In previous studies the authors demonstrated that DGKbeta induces neurite spine formation contributing to cognitive function but the exact mechanism it is not fully known. Now the authors follow up previous studies that demonstrated kinase independent function of DGKbeta and propose a mechanism that includes binding of RalA and subsequent activation of PLD and mTORC1.

The study although interesting and well executed the mechanistic insight is still lacking and some of the observations are inferred with no real demonstration of the sequence of events. Some of the major questions the authors need to address are:

1- In Figure 1 it would be important to show images with more than one cell. As it is shown spine growth in cells expressing DGKbeta is clear but the shape of cells expressing the kinase dead mutant looks very different. The authors showed previously that treatment with the DGK inhibitor prevents neurite formation. Does the inhibitor also affect cells expressing the kinase dead mutant? If so it could be that the inhibitor is altering interaction with some partner like RalA

2- The authors mention that they decided to investigate binding proteins and after “several trials “they found RalA as a DGKbeta interactor. It is not clear what type of approach they followed to identify interacting proteins.

  1. The pull-down experiments in Fig 2 are of very low quality. In Fig2A, looks like there is more protein pulled down from cells expressing the KN mutant with less GFP positive band. The authors mention that interaction is lower for the KN mutant but this should be better quantified. In Fig2B they show that mutants in the C1 domain interacts to less extent with RalA and conclude that interaction takes place through the C1 domain. However, the KN mutant conserves intact C1 domains and interaction is also lower so it is difficult to conclude that the interaction takes place through the C1 region. Better analysis should be performed comparing all the mutants in the same Blot and quantifying several experiments. Also using the inhibitor could provide additional insight (see 1)

  1. Colocalization experiments in Fig 3 should also be quantified. Please not that merge figure in the KN expressing cells is missing.

  1. Silencing experiments shown in Fig 4 confirm that RalA contributes to neurite outgrowth but does not provide any hint on the interaction with DGKbeta, this effect could be independent of any interaction.

  1. A similar argument stands for experiments with PLD and mTOR inhibitors. These experiments suggest that both PLD and mTOR have a function on spine growth but those mechanisms have already been discussed by the authors. Additional experiments should be performed to provide a better model of how DGKbeta contributes to PLD and/or mTOR regulated functions. In this regard data in Figure 6B (S6phosphorylation, should include western blot images to show the real effect on S6 phosphorylation and not the ratio to control value).

  1. The authors mention that RalA interacts with PLD, have they tried to search of DGKbeta interaction with RalA promotes this interaction? Is PLD present in the DGKbeta/RalA complex? Is the complex modulated by DGKbeta activity? This type of questions would provide a better model of why a PA producing enzyme like DGKbeta couples to a different PA producing enzyme like PLD and if this represent redundant mechanisms to sustain neurite growth.

The article needs careful editing of English language and style.

Author Response

Thank you for constructive comments to make the manuscript better. According to your suggestions, we performed additional experiments and revised manuscript.

The revised sentence was in red. The followings are our responses to respective reviewer’s comments.

Comments to reviewer #2

  1. I changed the fig 1 according to your advice. Indeed, the degree of neurite outgrowth induced by DGKbetaKN was weaker than that by WT. The DGK inhibitor didn’t affect expression level of kinase dead mutant. Instead, the inhibitor affects interaction between RalA. To show the result, we added Fig. 7.
  2. We revised precisely the description why we focused on RalA, according to your suggestion.
  3. We performed additional experiments and changed Fig. 2.
  4. Thank you for your comment. I changed the fig.
  5. You are right. So, we changed the title and abstract. In addition. we performed additional experiments to strengthen the involvement of RalA, and added Fig. 4C.
  6. We revised Fig.6B according to your suggestion.

Thank you very much for your constructive comment, kinase activity of DGKbeta indeed affected the interaction between RalA and DGKbeta as shown in Fig 2, although we didn’t have direct evidence the complex includes PLD and its amount was changed by kinase activity of DGKbeta.

Reviewer 3 Report

Kano et al previously published the DGKbeta kinase activity promotes neurite outgrowth and that this occurs by activating mTORC1.  In the current studies, they examined how DGKbeta could have a kinase-independent function in neurite outgrowth.  They examined the role of RalA, which binds to the C1 domain of DGKbeta.  First, they showed that expression of DGKb-KN led to neurite outgrowth, although this growth is not as robust compared to the WT-DGKbeta.  They then examined how RalA may bind to DGKb.  Their results indicate interaction specifically with the WT but mutation of the C1A domain of DGKbeta led to decreased RalA binding.  RalA and DGKbeta also co-localized at the plasma membrane whereas the C1 mutants did not specifically localize to the membrane, in neuroblastoma cells.  Knockdown of RalA then decreased neurite outgrowth of KN-expressing cells.  Since RalA is known to bind to PLD, they also examined how PLD inhibition using ethanol and FIPI could affect neurite outgrowth and found that they prevented neurite induction by KN.  Lastly, they found that mTORC1 inhibition by rapamycin could prevent neurite outgrowth induced by KN.  Based on these findings they concluded that “sequential activation of RalA, PLD and mTORC1 is important for kinase-independent pathwaty in the DGKbeta-induced neurite outgrowth”.

The effect of the DGKbeta-KN on neurite outgrowth is convincing.  The conclusion (title) of the study is however an overstatement and should be rephrased.  At best, the data only suggest that RalA and mTORC1 are involved in neurite outgrowth induced by DGKbeta-KN.  Since binding of DGKbeta-KN and RalA is not convincing, it is unclear whether RalA is directly mediating the effect of the KN.  They should also analyze effect of overexpressing both KN and RalA on neurite outgrowth and mTORC1 activation.

Author Response

Thank you for constructive comments to make the manuscript better. According to your suggestions, we performed additional experiments and revised manuscript.

The revised sentence was in red. The followings are our responses to respective reviewer’s comments.

Comments to reviewer #3

Thank you for your comments. According to your suggestion, we changed the title and abstract. In addition, we added and revised some figs and text according to comments by other reviewers.

Round 2

Reviewer 1 Report

I appreciate the effort authors put to improve the manuscript.

Just one thing- the authors misunderstood one of my review comments. "0.01 M PBS/TBS" (now lines 95 and 101) do not make sense. PBS is a generic name of buffers, that contain a certain concentration of phosphate, sodium, etc. "PBS" itself should not have concentration. We can only say 1x concentrated PBS, for example. The same is true for TBS.

Author Response

To reviewer #1,

Thank you very much for your kind comments.

I revised the description related PBS and TBS according to your suggestion in the final version.

Yasu

Yasuhito Shirai, Ph.D

Kobe Univ

Reviewer 2 Report

The authors have addressed all the issues. In my opinion the Ms is better focused and easier to understand. 

I would recomment some english editing

Author Response

To reviewer #2,

Thank you very much for your effort to finalize the manuscript.

The final version of the manuscript had English editing by nature publishing group.

Yasuhito Shirai, Ph.D